

# On mineral dust aerosol hygroscopicity
Lanxiadi Chen,[1,6] Chao Peng,[1] Athanasios Nenes,[2,5] Wenjun Gu,[1,6] Hanjing Fu,[3] Xing Jian,[3] Huanhuan Zhang,[1,6]
Guohua Zhang,[1] Jianxi Zhu,[4] Xinming Wang,[1,6,7] Mingjin Tang[1,6,7,*]
[1] State Key Laboratory of Organic Geochemistry, Guangdong Key Laboratory of Environmental Protection and
Resources Utilization, and Guangdong-Hong Kong-Macao Joint Laboratory for Environmental Pollution and
Control, Guangzhou Institute of Geochemistry, Chinese Academy of Sciences, Guangzhou 510640, China
[2] Civil and Environmental Engineering, École Polytechnique Fédérale de Lausanne (EPFL), Lausanne,
Switzerland
[3] State Key Laboratory of Marine Environmental Science, College of Ocean and Earth Sciences, Xiamen
University, Xiamen, China
[4] CAS Key Laboratory of Mineralogy and Metallogeny and Guangdong Provincial Key Laboratory of Mineral
Physics and Material Research and Development, Guangzhou Institute of Geochemistry, Chinese Academy of
Sciences, Guangzhou, China
[5] Institute of Chemical Engineering Sciences, Foundation for Research and Technology-Hellas, Patras, Greece
[6] University of Chinese Academy of Sciences, Beijing, China
[7] Center for Excellence in Regional Atmospheric Environment, Institute of Urban Environment, Chinese
Academy of Sciences, Xiamen, China
* Correspondence: Mingjin Tang (mingjintang@gig.ac.cn)





**Abstract**

Despite its importance, hygroscopicity of mineral dust aerosol remains highly uncertain.

In this work, we investigated water adsorption and hygroscopicity of different mineral dust
samples at 25 °C, via measuring sample mass at different relative humidity (RH, up to 90%) using
a very sensitive balance. Mineral dust samples examined (twenty one in total) included seven
authentic mineral dust samples from different regions in the world and fourteen major minerals
contained in mineral dust aerosol. At 90% RH, mass ratios of adsorbed water to the dry mineral
ranged from 0.0011 to 0.3080, largely depending on the BET surface areas of mineral dust samples.
The surface coverages of adsorbed water were determined to vary between 1.26 and 8.63 at 90%
RH, and it was found that the Frenkel-Halsey-Hill (FHH) adsorption isotherm could well describe
surface coverages of adsorbed water as a function of RH, with $A_{FHH}$ and $B_{FHH}$ parameters in the
range of 0.15-4.39 and 1.10-1.91, respectively. The comprehensive and robust data obtained would
largely improve our knowledge of hygroscopicity of mineral dust aerosol.





## 1 Introduction

Mineral dust aerosol mainly comes from arid and semi-arid areas (Ginoux et al., 2012), such as Saharan desert, Taklimakan desert, and etc. Its annual flux and atmospheric loadings are estimated to be ~2000 Tg yr$^{-1}$ and ~19 Tg (Textor et al., 2006; Huneeus et al., 2011), making mineral dust one of the most important aerosols in the troposphere. Mineral dust aerosol has significant impacts on atmospheric chemistry, climate and biogeochemical cycles (Knippertz and Stuut, 2014). It can alter the radiative forcing of the earth both directly (Balkanski et al., 2007; Huang et al., 2014; Di Biagio et al., 2017) and indirectly (Cziczo et al., 2013; Karydis et al., 2017). Mineral dust can also change the abundance of reactive trace gases as well as aerosol compositions via heterogeneous reactions (Usher et al., 2003; Dupart et al., 2012; He et al., 2014; Tang et al., 2017; Yu and Jang, 2019). Furthermore, the deposition of mineral dust will bring substantial amounts of nutrients (e.g., Fe and P) into some marine and terrestrial ecosystems, thereby largely affecting biogeochemistry in these regions (Jickells et al., 2005; Okin et al., 2011; Schulz et al., 2012; Li et al., 2017; Tagliabue et al., 2017; Meskhidze et al., 2019).

Hygroscopicity largely determines the impacts of mineral dust aerosol on atmospheric chemistry and climate. For examples, many studies found that relative humidity (RH) and thus the amount of water associated with mineral dust have profound effects on the rates, mechanisms and products of heterogeneous reactions (Vlasenko et al., 2009; Rubasinghege and Grassian, 2013; Tang et al., 2014; Tang et al., 2017; Lasne et al., 2018; Wang et al., 2018; Yu and Jang, 2018; Mitroo et al., 2019). In addition, hygroscopicity of mineral dust aerosol plays important roles in its optical properties (and thus the direct radiative effect) and its ability to act as cloud condensation nuclei and ice-nucleating particles (and thus the indirect radiative effect) (Sorjamaa and Laaksonen, 2007; Kumar et al., 2009; Garimella et al., 2014; Kreidenweis and Asa-Awuku, 2014; Laaksonen





et al., 2016; Tang et al., 2016; Tang et al., 2019a). Therefore, a number of previous studies have
investigated water adsorption and hygroscopic properties of mineral dust aerosol at <100% RH,
as reviewed by Tang et al. (2016). However, different studies displayed considerable discrepancies
as large as a few orders of magnitude (Tang et al., 2016), thus precluding a good understanding of
the roles mineral dust aerosol plays in atmospheric chemistry and climate.

As pointed out by Tang et al. (2016), such discrepancies are largely due to the non-

sphericity and low hygroscopicity of mineral dust particles, making it difficult to quantify the
amount of water associated with them at elevated RH. Instruments which measure mobility or
optical diameters of aerosol particles often found that the diameters of mineral dust particles did
not increase significantly (or even showed considerable decrease due to particle restructuring
during humidification) with increasing RH (Gustafsson et al., 2005; Vlasenko et al., 2005; Herich
et al., 2009; Koehler et al., 2009; Attwood and Greenslade, 2011). Fourier transform infrared
spectroscopy (FTIR) is a sensitive method to detect adsorbed water on mineral dust (Goodman et
al., 2001; Ma et al., 2010a; Joshi et al., 2017); however, it is not a trivial task to convert the intensity
of its infrared absorption to the amount of adsorbed water (Schuttlefield et al., 2007b; Ma et al.,
2010b; Tang et al., 2016). Quartz crystal microbalance (QCM) is another sensitive technique to
examine water adsorption and absorption (Schuttlefield et al., 2007b; Navea et al., 2010; Yeşilbaş
and Boily, 2016); however, it is in doubt that the underlying assumptions required to convert the
change in resonance frequency of the quartz crystal to the change in sample mass are always
fulfilled (Tang et al., 2016; Tang et al., 2019a).

In our previous work (Gu et al., 2017), we developed a new method to investigate

hygroscopic properties of atmospherically relevant particles using a vapor sorption analyzer, in
which a very sensitive balance was employed to measure the mass of a sample (typically with a



dry mass of tenths or a few mg) as different RH under isotherm conditions. Comprehensive
validation carried out confirmed the robustness of this method (Gu et al., 2017), and this instrument
has been employed to study hygroscopic properties of various particles, including nonspherical
particles such as saline mineral dust and pollen grains (Chen et al., 2019; Tang et al., 2019b; Tang
et al., 2019c). This instrument was used in the present work to quantitatively measure hygroscopic
properties of a number of mineral dust particles, including several authentic mineral dust samples
from different regions in the world and individual minerals commonly found in mineral dust
aerosol. We also attempted to figure out which theoretical models could describe hygroscopic
properties of mineral dust particles, and examined the dependence of mineral dust hygroscopicity
on several parameters (such as particle diameter, surface area and the mass fraction of soluble
materials).

## 2 Experimental section


### 2.1 Sample information


In total twenty one different types of mineral dust were investigated, including fourteen

major minerals commonly found in mineral dust aerosol (Formenti et al., 2011; Nickovic et al.,
2012; Journet et al., 2014; Scanza et al., 2015; Engelbrecht et al., 2016) and seven authentic
mineral dust samples, and their information can be found in Table 1. The fourteen major minerals
examined included four oxides ($SiO_2$, $TiO_2$, magnetite and hematite), one oxyhydroxide (goethite),
three feldspars (potassium feldspar, albite and microcline), two carbonates ($CaCO_3$ and dolomite)
and four clay minerals (montmorillonite, illite, kaolinite and chlorite). As shown in Table 1, $SiO_2$,
montmorillonite and kaolinite were supplied by Sigma Aldrich; $TiO_2$ (P25) was supplied by
Degussa; hematite and magnetite were supplied by Strem; goethite was provided by Santa Cruz;
microcline, $CaCO_3$ and dolomite were provided by Alfa Aesar. Potassium feldspar and albite were





obtained from National Research Center of Testing Techniques for Building Materials
(NRCTTBM, Beijing, China), and illite (IMt-1) was obtained from the Clay Mineral Society at
Purdue University, Indianan, USA (Schuttlefield et al., 2007b; Tang et al., 2014). In addition,
chlorite was collected by one co-author from Liaoning Province, China.

**Table 1.** Measured BET surface areas (BET), average particle diameters ($d_p$) and sources of
mineral dust samples examined in this work.

| sample | BET (m$^2$/g) | $d_p$ (µm) | source |
|---|---|---|---|
| SiO$_2$ | 6.54±0.01 | 1.65 | Sigma Aldrich |
| TiO$_2$ | 54.60±0.01 | 1.66 | Degussa |
| hematite | 9.23±0.17 | 0.80 | Strem |
| goethite | 13.41±0.01 | 1.00 | Santa Cruz |
| magnetite | 6.34±0.04 | 1.70 | Strem |
| potassium feldspar | 3.96±0.01 | 8.25 | NRCTTBM |
| albite | 3.62±0.02 | 5.51 | NRCTTBM |
| microcline | 2.17±0.01 | 14.33 | Alfa Aesar |
| CaCO$_3$ | 2.18±0.01 | 3.12 | Alfa Aesar |
| dolomite | 11.79±0.05 | 7.41 | Alfa Aesar |
| illite | 24.04±0.14 | 20.23 | The Clay Minerals Society |
| kaolinite | 9.64±0.01 | 9.99 | Sigma Aldrich |
| montmorillonite | 249.91±0.42 | 23.95 | Sigma Aldrich |
| chlorite | 9.95±0.03 | 19.19 | Liaoning, China |
| ATD | 36.67±1.06 | 1.05 | Powder Technology Inc. |
| China loess | 11.71±0.02 | 2.44 | Chinese Academy of Geological Science |
| QH dust | 8.79±0.02 | 18.56 | Chinese Academy of Geological Science |
| TLF dust | 8.49±0.01 | 8.04 | Turpan, Xinjiang, China |
| Bordj dust | 16.40±1.20 | 32.30 | M'Bour, Algeria |
| M'Bour dust | 14.50±1.00 | 54.41 | Bordj, Senegal |
| Saharan dust | 51.46±0.34 | 23.70 | Cape Verde |


The seven authentic mineral dust samples were obtained from Africa, Asia and North
America. As shown in Figure 1, three authentic mineral dust samples (M'Bour dust, Bordj dust
and Saharan dust) were collected from topsoil in Senegal, Algeria and Cape Verde Islands (Tang
et al., 2012; Joshi et al., 2017), respectively. QH dust (which is brown desert soil) and China loess,
collected from topsoil in Qinghai and Shaanxi, were supplied by Chinese Academy of Geological
Science as certificated materials (GBW07448 and GBW07454) (Tang et al., 2019c). TLF dust
were airborne dust particles collected on 23 April 2010 at an urban site in Turpan (Xinjiang, China)
during a major dust storm. In addition, Arizona Test Dust (ATD, nominal 0-3 µm fraction), an
authentic mineral dust sample commercially available from Powder Technology Inc. (Minnesota,
USA) and widely used in atmospheric aerosol research (Vlasenko et al., 2005; Sullivan et al.,
2010a; Tang et al., 2016), was also investigated in our work.

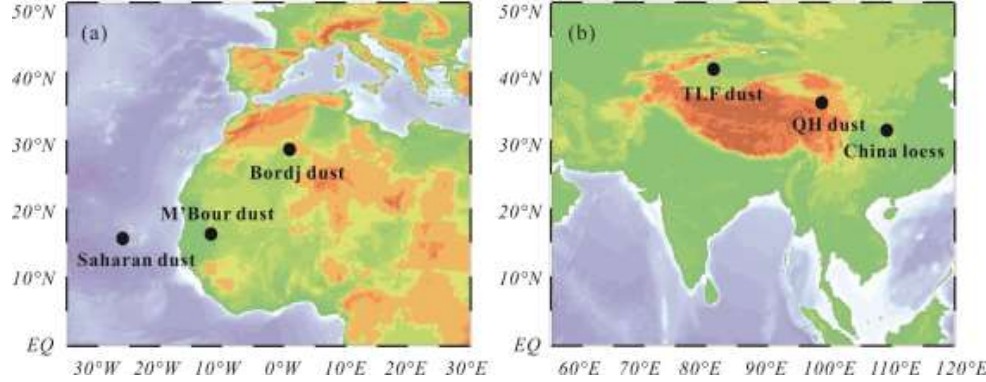


**Figure 1.** Locations where (a) African and (b) Asian authentic mineral dust samples examined in
this work were collected.

When received, three feldspars, dolomite, illite, chlorite and TLF dust contained significant
amounts of rock chips or giant particles; as a result, they were pretreated using the procedure
described in our previous work (Tang et al., 2019c). In brief, these samples were dried at 120 ºC



for 24 hours using an oven; after that, they were ground manually and then using a ball mill so that
most particles were <74 µm in diameter; finally, these samples were dried again at 120 °C for 24
hours and then cooled down. All the samples were stored in plastic bottles which were tightly
sealed to prevent contamination by lab air.
**2.2 Sample characterization**

Dynamic light scattering (JL-1177, Jingxin Powder Technologies Inc., Chengdu, Sichuan,

China) was employed to measure size distributions of mineral dust samples examined in our work.
In addition, Brunauer-Emmett-Teller (BET) surface areas of these samples were determined using
an accelerated surface area and porosimetry analyzer (ASAP 2020 PLUS, Micromeritics, Georgia,
USA), and $N_2$ was used as the adsorbate. Details on particle size and BET surface area
measurements can be found elsewhere (Li et al., 2020).

To measure their inorganic soluble compositions, each mineral dust sample (~10 mg) was

mixed with 10 mL ultrapure deionized water, and the mixture was stirred for 2 hours using an
oscillating table. After centrifugalization, the solution was filtered using a 5 mL syringe fitted with
a 0.2 µm PTFE membrane filter and then analyzed using ion chromatography (Metrohm model
761 Compact IC, Metrohm, Herisau, Switzerland). More information on ion chromatography
analysis can be found in our previous work (Tang et al., 2019c). We attempted to measure five
cations ($Na^+$, $K^+$, $NH_4^+$, $Mg^{2+}$ and $Ca^{2+}$) and seven anions ($NO_3^-$, $SO_4^{2-}$, $Cl^-$, $NO_2^-$, $Br^-$, $F^-$ and
$PO_4^{3-}$), and their detection limits were estimated to be around 0.02 mg/L.
**2.3 Hygroscopicity measurements**

Hygroscopic properties of mineral dust samples were investigated using a vapor sorption

analyzer (Q5000SA, TA instruments, Delaware, USA). This instrument, described in our previous
work (Gu et al., 2017; Chen et al., 2019; Tang et al., 2019b), measured sample mass as a function



of RH under isotherm conditions. Measurements could be conducted in the RH range of 0-98%
and in the temperature range of 5-85 $^{\circ}$C. We routinely measured the deliquescence RH of NaCl,
$(NH_4)_2SO_4$ and KCl at 25 $^{\circ}$C, and the measured values differed from the actual values by <1% RH.

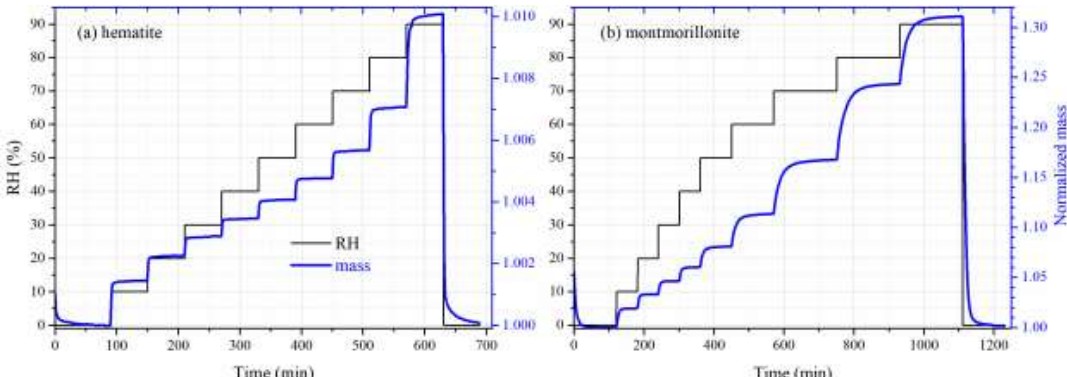

**Figure 2.** RH (black curve, left *y*-axis) and mass of mineral dust (normalized to that at <1% RH,
blue curve, right *y*-axis) as a function of experimental time: (a) hematite; (b) montmorillonite.

In this work, the initial masses of mineral dust samples used typically ranged from 5 to 15
mg. As displayed in Figure 2, the sample under investigation was first dried at <1% RH; after that,
RH was increased in a stepwise manner to 90%, and at each step RH was increased by 10%; at
last, the sample was dried again at <1% RH. At each step we changed the RH only after the samples
mass became stable, and the sample mass was considered to be stable when the mass change was
<0.05% in 30 min. All the experiments were carried out in triplicate at 25 $^{\circ}$C.
**3 Results**
**3.1 Sample characteristics**
As shown in Table 1, the BET surface areas were found to vary between 2.17±0.01
(microcline) and 249.91±0.42 m$^2$/g (montmorillonite), spanning over two orders of magnitude.
Except for montmorillonite, the BET surface areas were in the range of a few to tens of m$^2$/g. In



addition, the average particle diameters ($d_p$) were determined to range from 0.80 µm (hematite) to
54.41 µm (M'Bour dust), and their size distributions can be found in Figures S1-S7.
Tables S1-S2 show mass fractions of water soluble inorganic ions for the twenty one
mineral dust samples considered in this study. $Na^+$, $K^+$, $Ca^{2+}$, $Mg^{2+}$, $F^-$, $Cl^-$ and $SO_4^{2-}$ were detected
in most of the samples, while $NH_4^+$ was above its detection limit only for two samples. The total
mass fractions of all the soluble inorganic ions were found to be quite low, ranging from 0.16 mg/g
for $SiO_2$ and 12.55 mg/g for Bordj dust.

**3.2 Water uptake by different mineral dust**

As described in Section 2.3, sample mass of mineral dust was measured at different RH in
our work; therefore, the mass ratio of adsorbed water to the dry mineral, $m_w/m_0$, could then be
determined as a function of RH. Furthermore, $m_w/m_0$ could be converted to surface coverage of
adsorbed water ($\theta$), using Eq. (1) (Tang et al., 2016):
$$\theta = \frac{m_w}{m_0} \cdot \frac{N_A \cdot A_w}{M_w \cdot A_{BET}} \qquad (1),$$
where $N_A$ is Avogadro constant ($6.02 \times 10^{23}$ mol$^{-1}$), $M_w$ is the molar mass of water (18 g mol$^{-1}$), $A_w$
is the surface area each adsorbed water molecule would occupy (assumed to be $1 \times 10^{-15}$ cm$^{-2}$), and
$A_{BET}$ is the BET surface (in cm$^2$ g$^{-1}$) of the mineral dust under consideration. Tables 2-5 summarize
$m_w/m_0$ and $\theta$ as a function of RH for all the mineral dust examined in our work. Please note that
our previous work (Tang et al., 2019c) discussed water uptake by China loess and QH dust, and
these results are included here to compare with the other nineteen mineral dust samples.

**Table 2.** Mass ratios of adsorbed water to the dry mineral ($m_w/m_0$) and surface coverages of
adsorbed water ($\theta$) as a function of RH (%) for $SiO_2$, $TiO_2$, magnetite, hematite, goethite and
potassium feldspar.





| RH | SiO$_2$ | | TiO$_2$ | | hematite | |
|---|---|---|---|---|---|---|
| | $m_w/m_0$ | $\theta$ | $m_w/m_0$ | $\theta$ | $m_w/m_0$ | $\theta$ |
| 10 | 0.0005±0.0001 | 0.25±0.02 | 0.0031±0.0011 | 0.19±0.07 | 0.0014±0.0001 | 0.52±0.02 |
| 20 | 0.0008±0.0001 | 0.40±0.05 | 0.0054±0.0012 | 0.33±0.07 | 0.0022±0.0001 | 0.81±0.03 |
| 30 | 0.0011±0.0001 | 0.55±0.05 | 0.0072±0.0012 | 0.44±0.07 | 0.0029±0.0001 | 1.03±0.04 |
| 40 | 0.0014±0.0001 | 0.70±0.05 | 0.0089±0.0012 | 0.54±0.07 | 0.0034±0.0002 | 1.24±0.06 |
| 50 | 0.0017±0.0001 | 0.86±0.06 | 0.0108±0.0012 | 0.66±0.08 | 0.0040±0.0002 | 1.46±0.07 |
| 60 | 0.0020±0.0001 | 1.04±0.07 | 0.0135±0.0013 | 0.82±0.08 | 0.0047±0.0002 | 1.72±0.07 |
| 70 | 0.0026±0.0002 | 1.32±0.09 | 0.0168±0.0013 | 1.03±0.08 | 0.0057±0.0002 | 2.06±0.07 |
| 80 | 0.0035±0.0003 | 1.81±0.14 | 0.0218±0.0013 | 1.34±0.08 | 0.0071±0.0002 | 2.56±0.07 |
| 90 | 0.0058±0.0007 | 2.95±0.35 | 0.0355±0.0013 | 2.17±0.08 | 0.0101±0.0002 | 3.68±0.08 |
| RH | goethite | | magnetite | | potassium feldspar | |
| | $m_w/m_0$ | $\theta$ | $m_w/m_0$ | $\theta$ | $m_w/m_0$ | $\theta$ |
| 10 | 0.0013±0.0001 | 0.33±0.02 | 0.0005±0.0001 | 0.27±0.01 | 0.0006±0.0001 | 0.54±0.01 |
| 20 | 0.0022±0.0002 | 0.55±0.04 | 0.0007±0.0001 | 0.39±0.07 | 0.0010±0.0001 | 0.84±0.01 |
| 30 | 0.0029±0.0002 | 0.73±0.06 | 0.0010±0.0001 | 0.52±0.08 | 0.0015±0.0003 | 1.24±0.25 |
| 40 | 0.0037±0.0005 | 0.92±0.12 | 0.0012±0.0001 | 0.64±0.07 | 0.0017±0.0003 | 1.46±0.25 |
| 50 | 0.0044±0.0005 | 1.10±0.12 | 0.0015±0.0001 | 0.77±0.07 | 0.0020±0.0003 | 1.70±0.24 |
| 60 | 0.0052±0.0005 | 1.30±0.12 | 0.0018±0.0001 | 0.93±0.07 | 0.0023±0.0002 | 1.92±0.17 |
| 70 | 0.0061±0.0004 | 1.53±0.11 | 0.0022±0.0001 | 1.15±0.07 | 0.0027±0.0002 | 2.25±0.17 |
| 80 | 0.0075±0.0004 | 1.88±0.10 | 0.0029±0.0001 | 1.55±0.04 | 0.0035±0.0002 | 2.92±0.19 |
| 90 | 0.0124±0.0004 | 3.09±0.11 | 0.0052±0.0003 | 2.72±0.16 | 0.0056±0.0003 | 4.73±0.21 |



Below we discuss hygroscopicity of mineral dust investigated, and compare our measured

$m_w/m_0$ and $\theta$ with those reported in previous work. As our work directly measured mass change of

mineral dust due to water uptake, we prefer to compare $m_w/m_0$ when such values were also reported

in previous studies; otherwise, we then choose to compare $\theta$. As aerosol-based measurements are

usually not sensitive enough and also need the particle sphericity assumption (Tang et al., 2016),

we do not compare our results with those measurements.





### 3.2.1 SiO$_2$ and TiO$_2$


In our work $m_w/m_0$ was determined to be 0.0011, 0.0020 and 0.0058 for SiO$_2$ at 30%, 60%
and 90% RH, corresponding to $\theta$ of 0.55, 1.04 and 2.95, respectively. Figure 3a compares our work
with previous studies in which FTIR (Goodman et al., 2001; Ma et al., 2010a; Joshi et al., 2017)
and QCM (Schuttlefield et al., 2007a; Yeşilbaş and Boily, 2016) were used to measure water
uptake by SiO$_2$. At a given RH, $\theta$ values reported by the four previous studies (Goodman et al.,
2001; Schuttlefield et al., 2007a; Ma et al., 2010a; Joshi et al., 2017) were generally larger than
our work, and the difference usually did not exceed a factor of three. Furthermore, the differences
between our work and the four previous studies became smaller at higher RH. For example, at 80%
RH our measured $\theta$ was very close to those reported by Ma et al. (2010a) and Joshi et al. (2017),
and at 90% RH our measured $\theta$ was 20-30% larger than those reported by the two studies (Ma et
al., 2010a; Joshi et al., 2017). Yeşilbaş and Boily (2016) employed a QCM to investigate water
adsorption on quartz (0.3-14 μm), and $\theta$ was determined to be ~2300 at ~70% RH, almost three
orders of magnitude larger than these reported in our work and other previous studies (Goodman
et al., 2001; Schuttlefield et al., 2007a; Ma et al., 2010a; Joshi et al., 2017); therefore, the results
reported by Yeşilbaş and Boily (2016) are not included in Figure 3a.

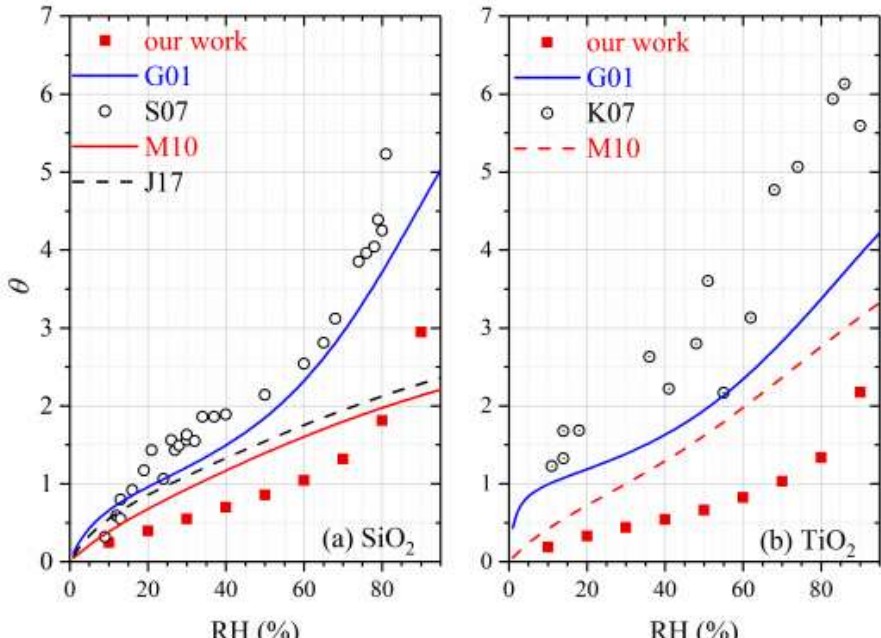


**Figure 3.** Comparison of surface coverages of adsorbed water ($\theta$) measured in our work with those

reported in previous studies for (a) $SiO_2$ and (b) $TiO_2$. G01: Goodman et al. (2001); S07:

Schuttlefield et al. (2007a); M10: Ma et al. (2010a); J17: Joshi et al. (2017); K07: Ketteler et al.

(2007).

For $TiO_2$, $m_w/m_0$ was determined to be 0.0072, 0.0135 and 0.0355 at 30%, 60% and 90%

RH, corresponding to $\theta$ of 0.44, 0.82 and 2.17, respectively. Water adsorption on P25 $TiO_2$ was

studied previously using FTIR (Goodman et al., 2001; Ma et al., 2010a), and another study

(Ketteler et al., 2007) employed atmospheric pressure X-ray photoelectron spectroscopy to explore

interactions of water vapor with the rutile single crystal surface (110). As shown in Figure 3b,

when compared with our work, $\theta$ values reported by Ma et al. (2010a) were higher across the entire

RH range, and the relative differences between our work and Ma et al. (2010a) were around a



factor of two or smaller. The relative differences between our work and the other two studies
(Goodman et al., 2001; Ketteler et al., 2007) were larger, being a factor of ~5 at lower RH and
becoming smaller at higher RH.

**3.2.2 Hematite, goethite and magnetite**

At 30%, 60% and 90% RH, $m_w/m_0$ was measured to be 0.0029, 0.0047 and 0.0101 for
hematite, corresponding to $\theta$ of 1.03, 1.72 and 3.68. Water adsorption on hematite was studied
previously using FTIR (Goodman et al., 2001; Ma et al., 2010a) and QCM (Yeşilbaş and Boily,
2016). Figure 4a reveals that our results agreed reasonably well with those reported by Goodman
et al. (2001) and Ma et al. (2010a), and the relative differences were found to be within a factor of
two. In addition, our results agreed fairly well with those reported for 10 nm hematite by Yeşilbaş
and Boily (2016), but were significantly smaller than their results for 50 nm hematite. Yeşilbaş
and Boily (2016) also studied water adsorption on 4 and 5 μm hematite particles, and $\theta$ were
reported to be ~300 at ~70% RH, almost two orders of magnitude larger than our results; therefore,
their measured $\theta$ for 4 and 5 μm hematite are not shown in Figure 4a.
In our work, $m_w/m_0$ was measured to be 0.0029, 0.0052 and 0.0124 at 30%, 60% and 90%
RH for goethite, corresponding to $\theta$ of 0.73, 1.30 and 3.09. Yeşilbaş and Boily (2016) employed
QCM to study water adsorption on goethite, and their measured $\theta$ are plotted in Figure 4b to
compare ours. Compared to our work, on average $\theta$ values measured by Yeşilbaş and Boily (2016)
were a factor of ~2 larger. We also investigated water adsorption on magnetite, and the results can
be found in Figure 4b. Compared to goethite, $\theta$ values were generally 20-30% smaller for
magnetite. As far as we know, water adsorption on magnetite was not quantitatively investigated
before.

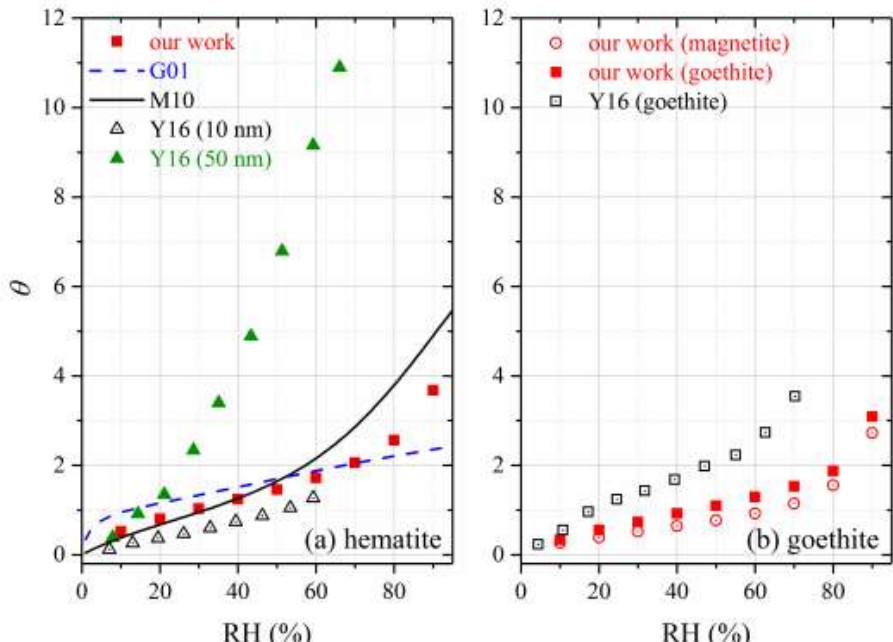

**Figure 4.** Comparison of surface coverages of adsorbed water ($\theta$) measured in our work with those reported in previous studies for (a) hematite and (b) goethite ($\theta$ measured in our work for magnetite are also plotted). G01: Goodman et al. (2001); M10: Ma et al. (2010a); Y16, Yeşilbaş and Boily (2016).

### 3.2.3 Feldspars

Tables 2-3 show that $m_w/m_0$ were determined to be 0.0056, 0.0060 and 0.0048 at 90% RH for potassium feldspar, albite and microcline, respectively; correspondingly, $\theta$ were found to be 4.73, 5.53 and 7.37. QCM was used by Yeşilbaş and Boily (2016) to study water uptake onto microcline, and $\theta$ was measured to be ~300 at ~70% RH, about two orders of magnitude larger than our results. We are not aware of other previous studies which investigated water adsorption on feldspars in a quantitative manner.




**Table 3.** Mass ratios of adsorbed water to the dry mineral ($m_w/m_0$) and surface coverages of

adsorbed water ($\theta$) as a function of RH (%) for albite, microcline, $CaCO_3$, dolomite, illite and

kaolinite.

| RH | albite | | microcline | | $CaCO_3$ | |
|---|---|---|---|---|---|---|
| | $m_w/m_0$ | $\theta$ | $m_w/m_0$ | $\theta$ | $m_w/m_0$ | $\theta$ |
| 10 | 0.0007±0.0002 | 0.67±0.20 | 0.0003±0.0001 | 0.51±0.10 | 0.0001±0.0001 | 0.10±0.07 |
| 20 | 0.0011±0.0002 | 1.00±0.19 | 0.0005±0.0001 | 0.81±0.14 | 0.0002±0.0002 | 0.27±0.24 |
| 30 | 0.0013±0.0001 | 1.19±0.04 | 0.0007±0.0001 | 1.06±0.19 | 0.0002±0.0002 | 0.38±0.28 |
| 40 | 0.0016±0.0001 | 1.45±0.04 | 0.0008±0.0002 | 1.28±0.26 | 0.0002±0.0001 | 0.33±0.17 |
| 50 | 0.0019±0.0001 | 1.74±0.04 | 0.0010±0.0002 | 1.57±0.29 | 0.0003±0.0001 | 0.41±0.22 |
| 60 | 0.0023±0.0001 | 2.10±0.03 | 0.0014±0.0002 | 2.11±0.30 | 0.0004±0.0002 | 0.63±0.31 |
| 70 | 0.0028±0.0001 | 2.63±0.05 | 0.0019±0.0001 | 2.96±0.22 | 0.0005±0.0002 | 0.79±0.34 |
| 80 | 0.0038±0.0001 | 3.50±0.06 | 0.0028±0.0002 | 4.40±0.33 | 0.0007±0.0003 | 1.02±0.39 |
| 90 | 0.0060±0.0001 | 5.53±0.06 | 0.0048±0.0006 | 7.37±0.98 | 0.0011±0.0005 | 1.73±0.79 |
| RH | dolomite | | illite | | kaolinite | |
| | $m_w/m_0$ | $\theta$ | $m_w/m_0$ | $\theta$ | $m_w/m_0$ | $\theta$ |
| 10 | 0.0004±0.0001 | 0.13±0.02 | 0.0050±0.0001 | 0.69±0.01 | 0.0014±0.0003 | 0.48±0.10 |
| 20 | 0.0007±0.0001 | 0.21±0.02 | 0.0083±0.0001 | 1.15±0.01 | 0.0024±0.0004 | 0.83±0.14 |
| 30 | 0.0009±0.0001 | 0.26±0.04 | 0.0110±0.0001 | 1.53±0.01 | 0.0032±0.0005 | 1.12±0.17 |
| 40 | 0.0011±0.0002 | 0.31±0.05 | 0.0135±0.0002 | 1.88±0.03 | 0.0040±0.0006 | 1.38±0.20 |
| 50 | 0.0013±0.0002 | 0.36±0.06 | 0.0157±0.0002 | 2.18±0.03 | 0.0047±0.0007 | 1.63±0.23 |
| 60 | 0.0015±0.0002 | 0.42±0.06 | 0.0181±0.0005 | 2.52±0.07 | 0.0056±0.0008 | 1.95±0.27 |
| 70 | 0.0018±0.0003 | 0.51±0.08 | 0.0210±0.0007 | 2.93±0.09 | 0.0070±0.0009 | 2.43±0.32 |
| 80 | 0.0025±0.0005 | 0.70±0.15 | 0.0253±0.0007 | 3.52±0.10 | 0.0093±0.0010 | 3.22±0.36 |
| 90 | 0.0045±0.0005 | 1.26±0.14 | 0.0333±0.0007 | 4.63±0.10 | 0.0146±0.0011 | 5.08±0.39 |



**3.2.4 Carbonates**





The mass ratio of adsorbed water to the dry mineral, $m_w/m_0$, was measured in our work to
be 0.0011 at 90% RH for $CaCO_3$, giving a $\theta$ value of 1.73. Water adsorption on $CaCO_3$ was
investigated previously, using thermogravimetric analysis (Gustafsson et al., 2005), physisorption
analysis (Ma et al., 2012a) and QCM (Hatch et al., 2008; Schuttlefield, 2008; Yeşilbaş and Boily,
2016). Hatch et al. (2008) and Ma et al. (2008) reported $m_w/m_0$ as a function of RH; Figure 5a
shows that compared to our work, $m_w/m_0$ values determined by Hatch et al. (2008) were
significantly larger (by a factor of 10 or more), whereas the results reported by Ma et al. (2012)
were smaller by a factor of ~2. We further compare our measured $\theta$ with those reported by another
two studies (Gustafsson et al., 2005; Schuttlefield, 2008). As shown in Figure 5b, the results
reported by Gustafsson et al. (2005) and Schuttlefield (2008) were found to be larger than ours, by
a factor of 2-3. In addition, $\theta$ was measured to be >100 at ~70% RH for $CaCO_3$ (Yeşilbaş and
Boily, 2016), approximately two orders of magnitude larger than our work.
As shown in Table 3, our work suggested that around 1.26 monolayers of adsorbed water
was formed on dolomite at 90% RH, similar to that for $CaCO_3$. To our knowledge, water
adsorption on dolomite has not been quantitatively explored by previous work.

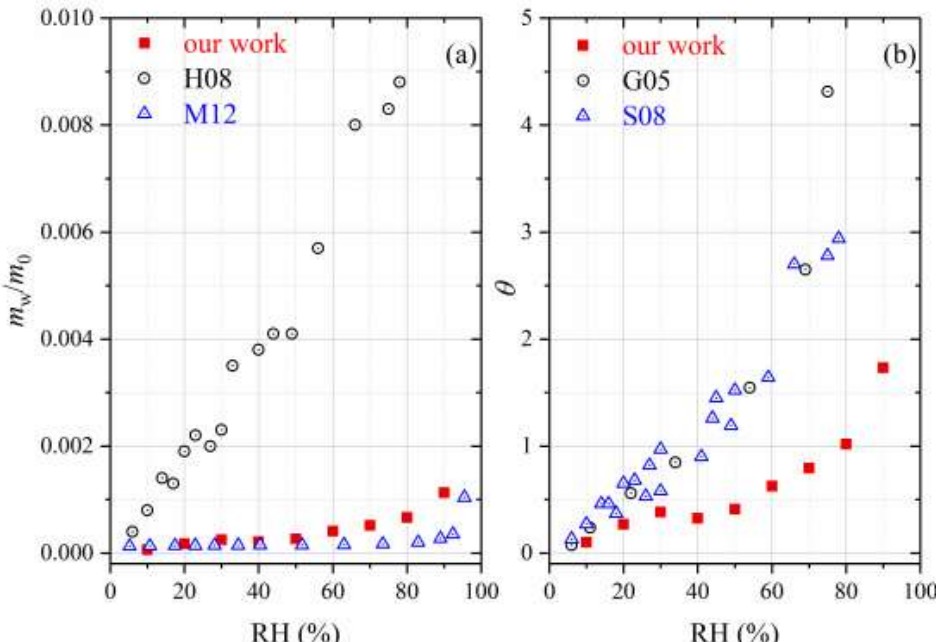


**Figure 5.** Comparison of water adsorption on $CaCO_3$ examined in different studies: (a) mass ratios

of adsorbed water to the dry mineral ($m_w/m_0$); (b) surface coverages of adsorbed water ($\theta$). G05,

Gustafsson et al. (2005); H08, Hatch et al. (2008); S08, Schuttlefield (2008); M12, Ma et al. (2012).


**3.2.5 Clay minerals**

For illite, $m_w/m_0$ and $\theta$ were determined to be 0.0333 and 4.63 at 90% RH in our study.

QCM was employed to study water adsorption on illite, and $m_w/m_0$ was reported to be 0.28 at ~90%

RH (Hatch et al., 2011) and ~0.27 at 75% RH (Schuttlefield et al., 2007b), around one order of

magnitude larger than our results. A recent study (Yeşilbaş and Boily, 2016) also investigated

water uptake onto illite using QCM, and their reported $\theta$ are compared with our results in Figure

6a. The relative differences between our and their (Yeşilbaş and Boily, 2016) work were usually

smaller than a factor of two, and became even smaller at higher RH.



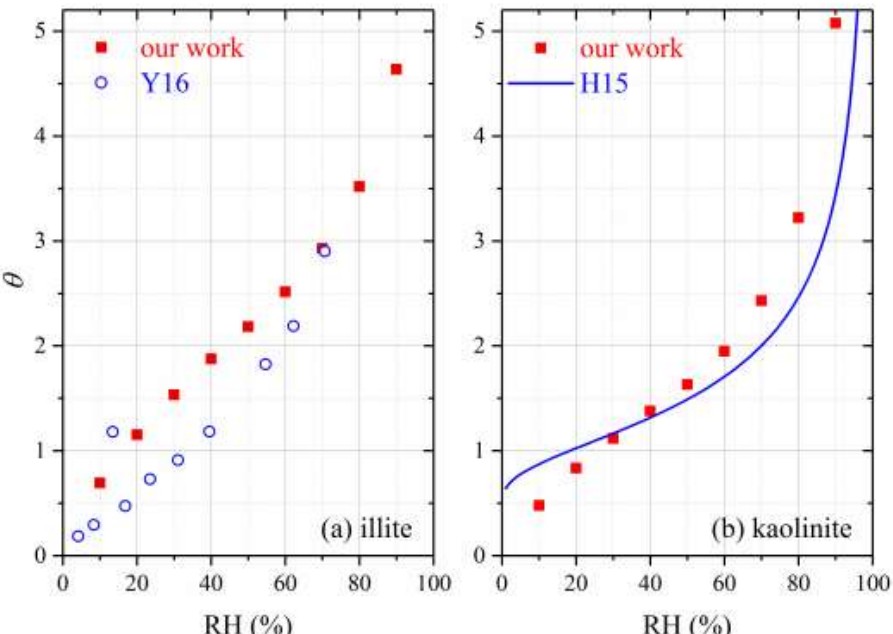

**Figure 6.** Comparison of surface coverages of adsorbed water ($\theta$) measured by different studies

for (a) illite and (b) kaolinite. H15, Hung et al. (2015); Y16, Yeşilbaş and Boily (2016).

For kaolinite, $m_w/m_0$ and $\theta$ were determined in our work to 0.0093 and 3.22 at 80% RH and

0.0146 and 5.08 at 90% RH, respectively. A few previous studies investigated water adsorption on

kaolinite using QCM (Schuttlefield et al., 2007b; Hatch et al., 2011; Yeşilbaş and Boily, 2016)

and physisorption analysis (Hung et al., 2015). Figure 6b compares our measured $\theta$ with those

reported by Hung et al. (2015), suggesting that the two studies were in good agreement, and the

relative differences were usually within 30%. At ~80% RH, $m_w/m_0$ were determined to be ~0.03

for kaolinite provided by Alfa and ~0.1 for kaolinite (KGa-1b) obtained from Clay Mineral Society

(Schuttlefield et al., 2007b), around three and ten times larger than our work. In the work by Hatch

et al. (2011), $m_w/m_0$ was determined to be ~0.1 at ~80% RH for kaolinite (KGa-1b), about one





order of magnitude larger than our result. Yeşilbaş and Boily (2016) examined water adsorption
on two different kaolinite samples (kaolinite provided by Fluka and KGa-1), and $\theta$ were found to
be up to 100 at ~70% RH, being >30 times larger than our work.

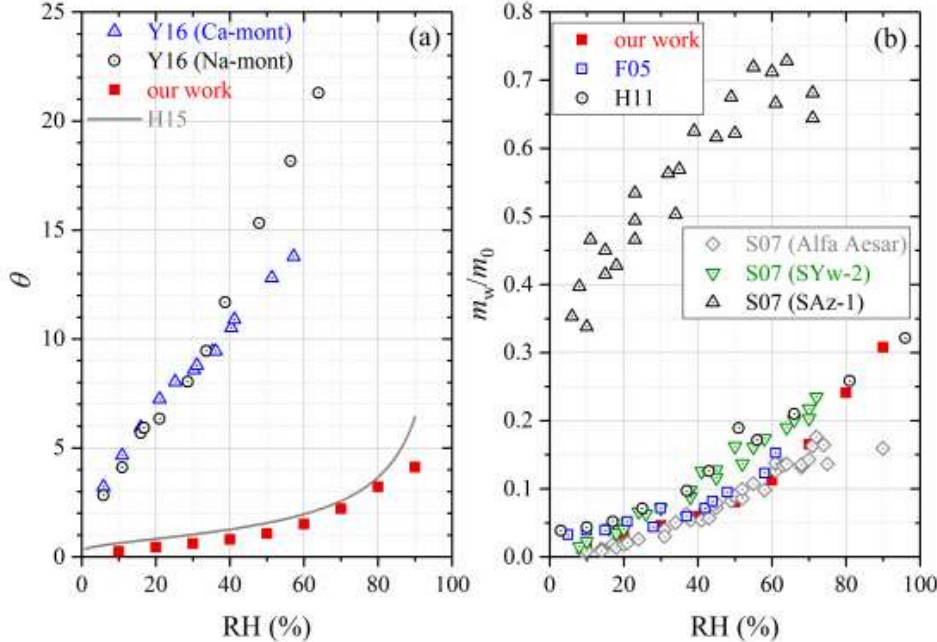


**Figure 7.** Comparison of water adsorption on montmorillonite examined in different studies: (a)
surface coverages of adsorbed water ($\theta$); (b) the mass ratio of adsorbed water to the dry mineral
($m_w/m_0$). F05, Frinak et al. (2005); S07, Schuttlefield et al. (2007); H11, Hatch et al. (2011); H15,
Hung et al. (2015); Y16, Yeşilbaş and Boily (2016).

We also studied water adsorption on montmorillonite, and $m_w/m_0$ and $\theta$ were measured to

be 0.308 and 4.12 at 90% RH. Physisorption analysis (Hung et al., 2015) and QCM (Yeşilbaş and
Boily, 2016) were utilized to investigate water uptake onto montmorillonite. As shown in Figure
7a, our work agreed well with Hung et al. (2015), and the results obtained by Yeşilbaş and Boily





(2016) for Ca- and Na-montmorillonite were much larger (by a factor of >10), when compared
with our work. Figure 7b compares our measured $m_w/m_0$ with those reported in previous studies in
which FTIR (Frinak et al., 2005) and QCM (Schuttlefield et al., 2007b; Hatch et al., 2011) were
used. In general good agreement between our work and the three previous studies were found,
except for SAz-1 montmorillonite (Schuttlefield et al., 2007b). One possible explanation for the
observed discrepancy is that montmorillonite samples from different sources may have different
hygroscopic properties. We note that prior to 2005, a few studies (Hall and Astill, 1989; Cases et
al., 1992; Xu et al., 2000; Zent et al., 2001) also investigated water uptake by montmorillonite, and
it was found that these studies agreed well with Frinak et al. (2005); therefore, the four studies
conducted before 2005 should also be consistent with our work.

In addition, water uptake by chlorite was explored in our work. As shown in Table 4, $m_w/m_0$

and $\theta$ were measured to be 0.012 and 4.03 at 90% RH. To our knowledge, hygroscopic properties
of chlorite have not been examined before.

**Table 4.** Mass ratio of adsorbed water to the dry mineral ($m_w/m_0$) and surface coverages of
adsorbed water ($\theta$) as a function of RH (%) for montmorillonite, chlorite, ATD, M'Bour dust,
Bordj dust and Saharan dust.

| RH | montmorillonite | | chlorite | | ATD | |
|---|---|---|---|---|---|---|
| | $m_w/m_0$ | $\theta$ | $m_w/m_0$ | $\theta$ | $m_w/m_0$ | $\theta$ |
| 10 | 0.0192±0.0002 | 0.26±0.01 | 0.0013±0.0001 | 0.42±0.02 | 0.0099±0.0001 | 0.90±0.01 |
| 20 | 0.0333±0.0003 | 0.45±0.01 | 0.0021±0.0001 | 0.70±0.03 | 0.0161±0.0002 | 1.47±0.02 |
| 30 | 0.0463±0.0004 | 0.62±0.01 | 0.0028±0.0001 | 0.94±0.04 | 0.0209±0.0001 | 1.91±0.01 |
| 40 | 0.0597±0.0008 | 0.80±0.01 | 0.0034±0.0001 | 1.14±0.04 | 0.0253±0.0001 | 2.31±0.01 |
| 50 | 0.0802±0.0009 | 1.07±0.01 | 0.0040±0.0001 | 1.34±0.05 | 0.0296±0.0001 | 2.70±0.01 |
| 60 | 0.1125±0.0011 | 1.51±0.02 | 0.0047±0.0001 | 1.57±0.05 | 0.0341±0.0001 | 3.11±0.01 |
| 70 | 0.1654±0.0023 | 2.21±0.03 | 0.0057±0.0001 | 1.93±0.05 | 0.0394±0.0001 | 3.59±0.01 |





| RH | M'Bour dust | | Bordj dust | | Saharan dust | |
|---|---|---|---|---|---|---|
| | 0.2407±0.0025 | 3.22±0.03 | 0.0077±0.0001 | 2.57±0.05 | 0.0470±0.0003 | 4.29±0.03 |
| | 0.3080±0.0029 | 4.12±0.04 | 0.0120±0.0002 | 4.03±0.08 | 0.0644±0.0009 | 5.87±0.08 |
| | $m_w/m_0$ | $\theta$ | $m_w/m_0$ | $\theta$ | $m_w/m_0$ | $\theta$ |
| 10 | 0.0014±0.0001 | 0.31±0.01 | 0.0010±0.0001 | 0.21±0.01 | 0.0102±0.0002 | 0.66±0.02 |
| 20 | 0.0023±0.0001 | 0.54±0.01 | 0.0020±0.0005 | 0.41±0.09 | 0.0166±0.0004 | 1.02±0.03 |
| 30 | 0.0032±0.0001 | 0.73±0.02 | 0.0026±0.0005 | 0.53±0.09 | 0.0214±0.0002 | 1.39±0.01 |
| 40 | 0.0039±0.0003 | 0.90±0.07 | 0.0034±0.0004 | 0.69±0.08 | 0.0260±0.0002 | 1.69±0.01 |
| 50 | 0.0046±0.0004 | 1.06±0.10 | 0.0040±0.0004 | 0.82±0.08 | 0.0304±0.0003 | 1.98±0.02 |
| 60 | 0.0052±0.0005 | 1.21±0.13 | 0.0050±0.0004 | 1.02±0.09 | 0.0360±0.0002 | 2.34±0.02 |
| 70 | 0.0069±0.0006 | 1.59±0.13 | 0.0076±0.0005 | 1.55±0.10 | 0.0438±0.0003 | 2.84±0.02 |
| 80 | 0.0092±0.0006 | 2.13±0.14 | 0.0118±0.0004 | 2.41±0.08 | 0.0557±0.0007 | 3.62±0.05 |
| 90 | 0.0152±0.0005 | 3.51±0.11 | 0.0192±0.0003 | 3.91±0.07 | 0.0793±0.0015 | 5.15±0.10 |


### 3.2.6 Authentic mineral dust

**ATD:** Table 4 suggests that at 90% RH, $m_w/m_0$ and $\theta$ were measured in our work to be 0.0644 and 5.87 for ATD. Two previous studies (Navea et al., 2010; Yeşilbaş and Boily, 2016) employed QCM to investigate water adsorption on ATD. In the first study (Navea et al., 2010), $m_w/m_0$ was measured to be >0.1 at 70% RH, being 2-3 times larger than our result (~0.04 at 70% RH); in the second study (Yeşilbaş and Boily, 2016), $\theta$ was measured to be >200 at ~70% RH, almost two orders of magnitude larger than our work (~3.6 at 70% RH). Gustafsson et al. (2005) used a thermogravimetric analyzer to study water uptake by ATD, and as shown in Figure 8a, their results agreed very well with ours. A recent study (Joshi et al., 2017) investigated water adsorption on ATD using FTIR; compared to our work, the values reported by Joshi et al. (2017) were ~30% lower, suggesting fairly good agreement between the two studies.



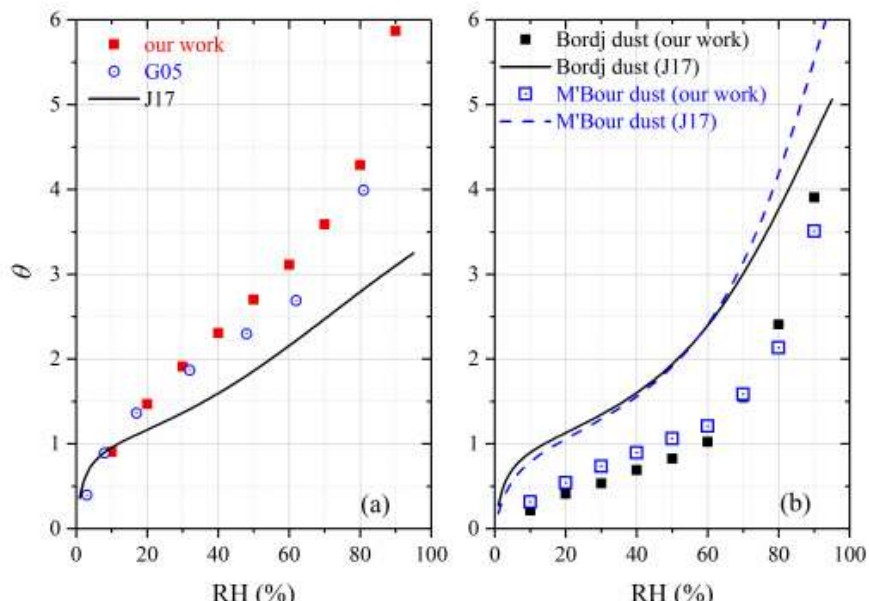

**Figure 8.** Comparison of surface coverages of adsorbed water ($\theta$) reported in different studies for

(a) ATD and (b) Bordj dust and M'Bour dust. G05, Gustafsson et al. (2005); J17, Joshi et al. (2017).

**African dust:** In our study, $m_w/m_0$ and $\theta$ were measured to be 0.0192 and 3.91 for Bordj

dust and 0.0152 and 3.51 for M'Bour dust at 90% RH. Joshi et al. (2017) employed FTIR to

investigate interaction of water vapor with Bordj dust and M'Bour dust. As suggested by Figure

8b, the relative differences between our and their work (Joshi et al., 2017) were usually within a

factor of two for the two dust samples, and the discrepancy also became smaller at higher RH,

suggesting fair consistence between the two studies.

For Saharan dust, $m_w/m_0$ and $\theta$ were determined in our study to be 0.0793 and 5.15 at 90%

RH. Water uptake onto Saharan dust was studied using QCM (Navea et al., 2010), and their results,

as shown in Figure 9a, were 2-3 times larger than our work.



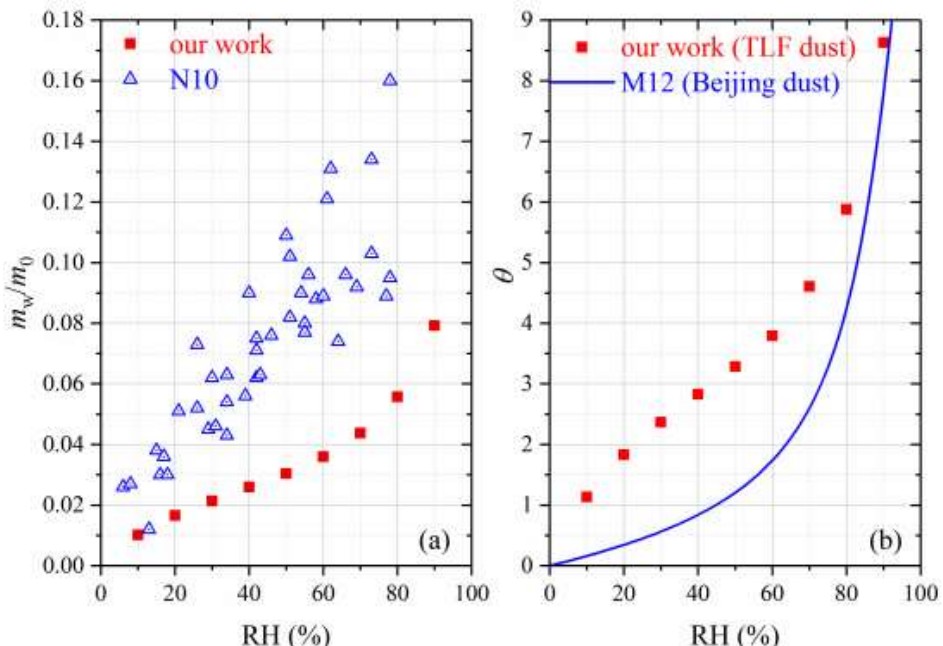

**Figure 9.** (a) Comparison of mass ratios of adsorbed water to the dry mineral ($m_w/m_0$) measured

by our work and N10 (Navea et al., 2010) for Saharan dust. (b) Comparison of surface coverages

of adsorbed water ($\theta$) for TLF dust measured in our work with Beijing dust measured by M12 (Ma

et al., 2012b).

**Asian dust:** Table 5 summarizes our results obtained for three Asian mineral dust samples,

including China loess, QH dust and TLF dust. It should be pointed out that $m_w/m_0$ have been

reported in our previous work (Tang et al., 2019c) for China loess and QH dust, and they are

included here for comparison. As shown in Table 5, the three Asian authentic dust samples

exhibited very similar water uptake properties, and their $m_w/m_0$ were determined to be 0.021-0.022

at 90% RH. Navea et al. (2010) employed QCM to study interaction of water vapor with China





loess, and $m_w/m_0$ was reported to be ~0.17 at 70% RH, more than one order of magnitude larger
than our result (~0.012 at 70% RH).

As mentioned in Section 2.1, TLF dust examined in our work were airborne dust particles

collected during a dust storm in Turpan (Xinjiang, China) which was very close to the dust source.
In a previous study (Ma et al., 2012b), dust particles (termed as Beijing dust here) were collected
during a dust storm in Beijing (and thus these particles had undergone atmospheric aging to some
extent), and their hygroscopic properties were then investigated using a physisorption analyzer.
As revealed by Figure 9b, our work agreed fairly well with Ma et al. (2012b) at high RH (70%,
80% and 90%), though the differences were considerably larger at lower RH.

**Table 5.** Mass ratios of adsorbed water to the dry mineral ($m_w/m_0$) and surface coverages of
adsorbed water ($\theta$) as a function of RH (%) for QH dust, China loess and TLF dust.

| RH | QH dust | | China loess | | TLF dust | |
|---|---|---|---|---|---|---|
| | $m_w/m_0$ | $\theta$ | $m_w/m_0$ | $\theta$ | $m_w/m_0$ | $\theta$ |
| 10 | 0.0022±0.0001 | 0.84±0.01 | 0.0030±0.0001 | 0.87±0.03 | 0.0029±0.0001 | 1.13±0.05 |
| 20 | 0.0037±0.0001 | 1.39±0.01 | 0.0049±0.0001 | 1.39±0.04 | 0.0047±0.0002 | 1.83±0.08 |
| 30 | 0.0049±0.0001 | 1.86±0.01 | 0.0062±0.0001 | 1.78±0.04 | 0.0060±0.0002 | 2.37±0.09 |
| 40 | 0.0060±0.0001 | 2.29±0.01 | 0.0074±0.0001 | 2.12±0.03 | 0.0072±0.0002 | 2.83±0.09 |
| 50 | 0.0072±0.0001 | 2.75±0.01 | 0.0087±0.0002 | 2.49±0.04 | 0.0083±0.0002 | 3.28±0.09 |
| 60 | 0.0086±0.0001 | 3.29±0.01 | 0.0102±0.0002 | 2.90±0.04 | 0.0096±0.0003 | 3.79±0.11 |
| 70 | 0.0104±0.0001 | 3.96±0.01 | 0.0119±0.0002 | 3.41±0.05 | 0.0117±0.0004 | 4.61±0.16 |
| 80 | 0.0134±0.0001 | 5.09±0.02 | 0.0146±0.0002 | 4.17±0.05 | 0.0149±0.0007 | 5.88±0.26 |
| 90 | 0.0215±0.0001 | 8.20±0.01 | 0.0212±0.0003 | 6.05±0.07 | 0.0219±0.0012 | 8.63±0.46 |


**3.2.7 Discussion**

To investigate water adsorption by mineral dust, one previous study (Gustafsson et al.,

2005) employed thermogravimetric analysis which measured sample mass as a function of RH





(essentially the same to VSA used in our study), and another two groups (Ma et al., 2012a; Ma et
al., 2012b; Hung et al., 2015) employed physisorption analysis which measured change in water
vapor pressure caused by adsorption onto mineral dust (Ma et al., 2010b). Thermogravimetric
analysis, physisorption analysis and the VSA technique used in our work can be considered as
absolutely quantitative, and as discussed in Section 3.2, in general our work agreed well with the
four previous studies (Gustafsson et al., 2005; Ma et al., 2012a; Ma et al., 2012b; Hung et al.,

2015).

FTIR was widely employed in previous work (Goodman et al., 2001; Frinak et al., 2005;

Ma et al., 2010a; Joshi et al., 2017; Ibrahim et al., 2018) to study water uptake onto mineral dust,
although it is not straightforward to convert IR absorption intensities of adsorbed water to its
absolute amounts (Schuttlefield et al., 2007a; Ma et al., 2010b; Tang et al., 2019a). The relative
differences between these studies and our work were typically within a factor of 2-3; since even
for dust samples with the same name, samples examined in different studies may actually differ
substantially in composition and water uptake properties, the agreement between these studies and
our work can be considered as fairly good.

QCM is another technique widely used to investigate water uptake onto mineral dust

(Schuttlefield et al., 2007b; Hatch et al., 2008; Schuttlefield, 2008; Navea et al., 2010; Hatch et al.,
2011; Yeşilbaş and Boily, 2016). As shown in Section 3.2, though good agreement was found for
some mineral dust between our work and these QCM studies, large discrepancies (up to 2-3 orders
of magnitude) were frequently observed. This implies that the underlying assumptions required to
convert the change in resonance frequency of the quartz crystal to the change in sample mass may
not always be fulfilled, and as a result the QCM results should be used with cautions.





For the same dust (at least with the same name), different samples with distinctive
hygroscopicity may have been used in our work and previous studies, contributing to the observed
discrepancies. To further understand and resolve the discrepancies identified, it will be very useful
to distribute the same samples to different groups (in which various techniques would be applied
to study their hygroscopic properties) and compare the results obtained. Similar strategies have
already been adopted before to compare different instruments used for ice nucleation research and
shown to be valuable (Hiranuma et al., 2015; DeMott et al., 2018).
**3.3 Hygroscopicity parameterizations**
It has been suggested that water adsorption and hygroscopicity of insoluble particles can
be parameterized as a function of RH by several theoretical models, including 1) the Brunauer-
Emmet-Teller (BET) adsorption isotherm (Goodman et al., 2001; Ma et al., 2010a; Joshi et al.,
2017; Ibrahim et al., 2018), 2) the Freundlich adsorption isotherm (Hatch et al., 2011), 3) Frenkel-
Halsey-Hill (FHH) adsorption isotherm (Kumar et al., 2011b; Hatch et al., 2014; Hung et al., 2015;
Hatch et al., 2019) and 4) the $\kappa$-Köhler equation (Chen et al., 2019; Tang et al., 2019b). In this
work we attempted to use the aforementioned four models to fit our experimental data. As shown
in Figure 10 (where $SiO_2$, albite, kaolinite and TLF dust are used as examples), our work suggested
that the FHH adsorption isotherm could well describe the measured hygroscopicity of mineral dust
samples as a function of RH. In addition, we found that the other three parameterization methods
could not fit our experimental data.

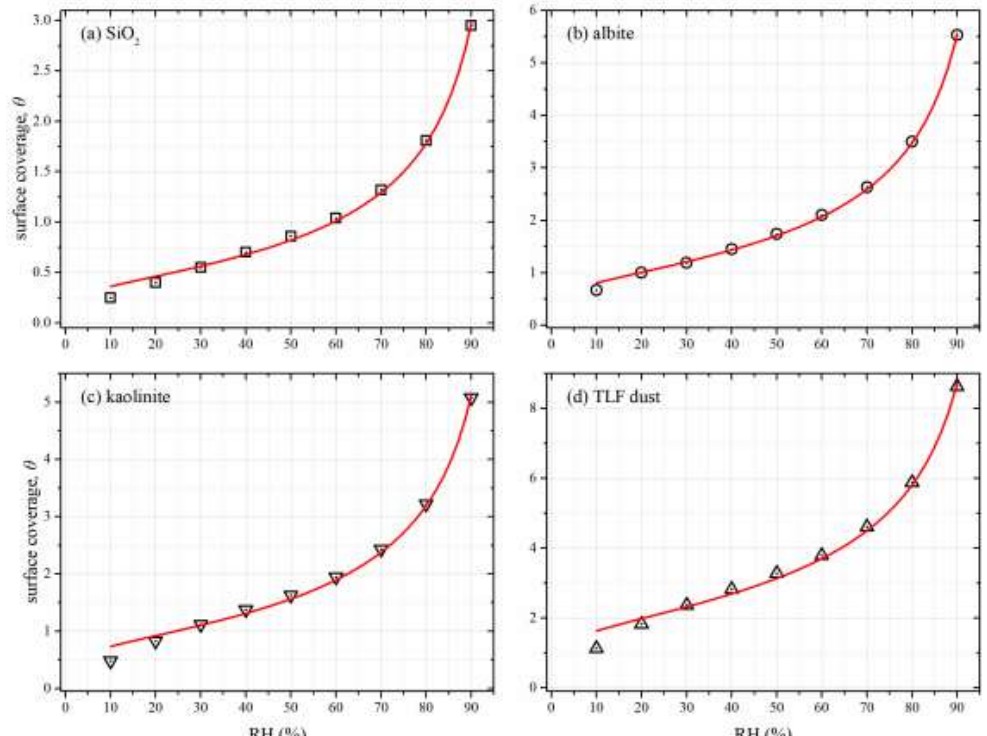


**Figure 10.** Surface coverages ($\theta$) of adsorbed water on (a) SiO$_2$, (b) albite, (c) kaolinite and (d)

TLF dust as a function of RH (0-90%) at 25 $^{\circ}$C. The experimental data were fitted with Frenkel-

Halsey-Hill adsorption isotherm model (solid curves).

The FHH adsorption isotherm, which describes surface coverages of adsorbed water ($\theta$) as

a function of RH, is given by Eq. (2) (Sorjamaa and Laaksonen, 2007; Tang et al., 2016):

$$\theta = \sqrt[B_{FHH}]{\frac{A_{FHH}}{-\ln(RH)}} \qquad (2),$$

where $A_{FHH}$ and $B_{FHH}$ are empirical parameters. We found that Eq. (2) can well fit $\theta$ versus RH for

all the 21 mineral dust samples examined, and the generated $A_{FHH}$ and $B_{FHH}$ values are summarized

in Table 6. As shown in Table 6, $A_{FHH}$ values spanned from 0.15±0.01 (dolomite) to 4.39±0.81

(ATD), while the variation of $B_{FHH}$ was much smaller, ranging from 1.10±0.04 (for Bordj dust) to





1.91±0.18 (for ATD). Our results were largely consistent with the theoretical work by Sorjamaa
and Laaksonen (2007), who suggested from a theoretical view that typical $A_{FHH}$ and $B_{FHH}$ values
should be in the range of 0.1-3.0 and 0.5-3.0.

**Table 6.** Comparison $A_{FHH}$ and $B_{FHH}$ values determined in our work for mineral dust with those
reported in previous studies. a: Kumar et al. (2011a); b: Hung et al. (2015); c: Hatch et al. (2019).

| sample | $A_{FHH}$ | $B_{FHH}$ | sample | $A_{FHH}$ | $B_{FHH}$ |
|---|---|---|---|---|---|
| TiO$_2$ | 0.35±0.01 | 1.52±0.05 | SiO$_2$ | 0.50±0.03 | 1.23±0.07 |
| hematite | 1.03±0.09 | 1.67±0.09 | | 2.95±0.05 [a] | 1.36±0.03 [a] |
| magnetite | 0.41±0.01 | 1.33±0.03 | CaCO$_3$ | 0.23±0.02 | 1.18±0.09 |
| goethite | 0.59±0.04 | 1.49±0.07 | | 3.00±0.04 [a] | 1.30±0.03 [a] |
| dolomite | 0.15±0.01 | 1.43±0.07 | illite | 1.96±0.23 | 1.56±0.21 |
| albite | 1.68±0.02 | 1.61±0.01 | | 1.02±0.38 [a] | 1.12±0.04 [a] |
| potassium feldspar | 1.10±0.06 | 1.42±0.09 | | 2.06 [c] | 2.19 [c] |
| microcline | 1.22±0.05 | 1.17±0.03 | kaolinite | 1.24±0.10 | 1.48±0.08 |
| chlorite | 0.96±0.06 | 1.55±0.07 | | 1.70 [b] | 2.25 [b] |
| China loess | 3.19±0.47 | 1.84±0.12 | montmorillonite | 0.65±0.05 | 1.13±0.07 |
| QH dust | 2.53±0.32 | 1.49±0.08 | | 2.06±0.72 [a] | 1.23±0.04 [a] |
| TLF dust | 4.08±0.60 | 1.59±0.12 | | 1.23±0.31 [a] | 1.08±0.03 [a] |
| Bordj dust | 0.49±0.03 | 1.10±0.04 | | 1.25 [b] | 1.33 [b] |
| M'Bour dust | 0.59±0.05 | 1.27±0.09 | | 2.28 [c] | 1.45 [c] |
| Saharan dust | 2.03±0.18 | 1.67±0.11 | ATD | 4.39±0.81 | 1.91±0.18 |
| | | | | 2.96±0.03 [a] | 1.28±0.03 [a] |


A few previous studies investigated hygroscopic properties (Hung et al., 2015; Hatch et al.,
2019) and CCN activities (Kumar et al., 2011b) of mineral dust, and reported $A_{FHH}$ and $B_{FHH}$ values
for samples they examined. Their results are also compiled in Table 6. As revealed by Table 6,
$B_{FHH}$ values reported in our work were reasonably consistent with previous studies, while larger
differences were observed for $A_{FHH}$ values. Another study (Kumar et al., 2011a) reported $A_{FHH}$ and





$B_{FHH}$ values for wet-generated mineral dust aerosols. Since the hygroscopicity of wet-generated
mineral dust aerosols could be very different from dry-generated aerosols (Sullivan et al., 2010b;
Kumar et al., 2011a), the results reported by Kumar et al. (2011b) for wet-generated aerosols are
not further discussed.
**4 Discussions**

As shown in Tables 2-5, among the 21 mineral dust samples examined, $m_w(90\%)/m_0$ (mass

ratios of adsorbed water at 90% RH to the dry sample) was found to range from 0.0011 for $CaCO_3$
to 0.0380 for montmorillonite, and $\theta(90\%)$ (surface coverages of adsorbed water at 90% RH)
varied between 1.26 for dolomite and 8.63 for TLF dust. It appears that clay minerals and authentic
mineral dust samples usually exhibited larger hygroscopicity on a per mass basis, when compared
to other mineral dust samples. $TiO_2$, for which $m_w(90\%)/m_0$ was only lower than ATD, Saharan
dust and montmorillonite, was an exception, probably because of its very large BET surface area
(54.6 $m^2$ $g^{-1}$).

One may expect that on a per mass basis, mineral dust samples with larger surface area would

have larger capacities to adsorb water. This was supported by our results shown in Figure 11a,
which suggests that for mineral dust samples considered in our study, overall $m_w(90\%)/m_0$
increased with the BET surface area. Nevertheless, not all the samples obeyed this general trend,
indicating that other factors would also play some roles in determining the ability of mineral dust
to adsorb water on a per mass basis. We also explored if there was any relationship between
hygroscopicity of mineral dust samples and soluble materials they contained. It was found that for
the 21 mineral dust samples considered in our work, $m_w(90\%)/m_0$ did not show any overall
dependence on the amounts of soluble inorganic ions.

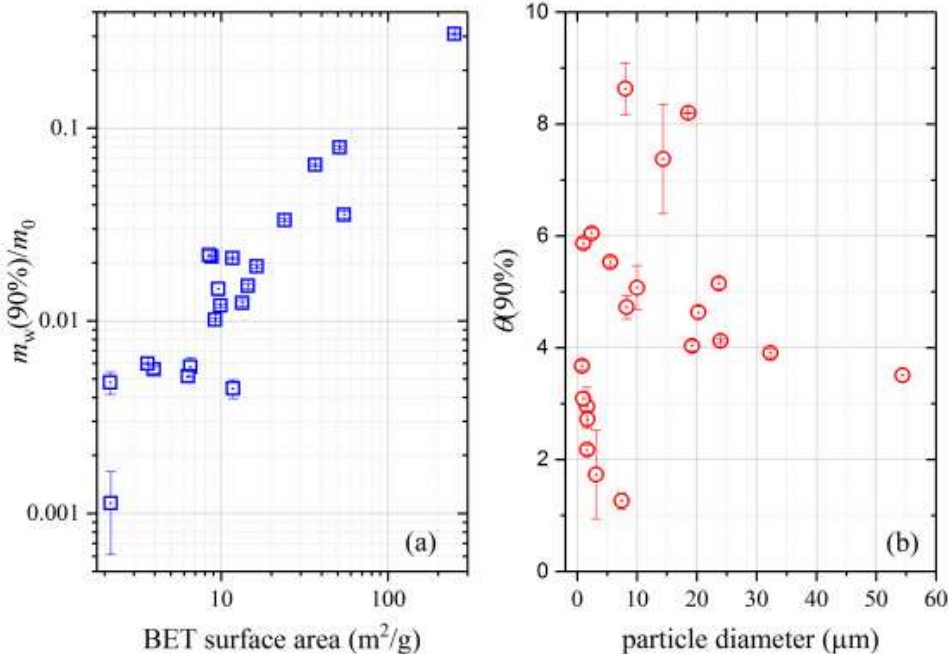


**Figure 11.** (a) The dependence of $m_w(90\%)/m_0$ (mass ratios of adsorbed water to the dry mineral

at 90% RH) on BET surface areas; (b) the dependence of $\theta(90\%)$ (surface coverages of adsorbed
water at 90% RH) on average particle diameters.

Ibrahim et al. (2018) studied water adsorption on ATD particles with different particle sizes,

and found that the RH at which one monolayer of adsorbed water was formed increased with
particle size; in other words, at the same RH the surface coverages of adsorbed water would be
higher for smaller particles (Ibrahim et al., 2018). In contrast, Yeşilbaş and Boily (2016)
investigated water adsorption on different mineral samples (21 in total), and suggested that at the
same RH more monolayers of adsorbed water would be formed on larger particles. However, as
shown in Figure 4b, our work revealed that surface coverages of adsorbed water at 90% RH
showed no dependence on particle size for the 21 mineral dust samples examined in our work.



## 5 Conclusions

Hygroscopicity largely determines environmental and climatic effects of mineral dust aerosol, one of the most abundant tropospheric aerosols. However, hygroscopic properties of mineral dust remain highly uncertain, due to relatively low hygroscopicity of mineral dust and its non-sphericity. In our work, a vapor sorption analyzer, which measured sample mass as a function of RH, was employed to investigate water adsorption and hygroscopic properties of 21 different mineral dust samples, including seven authentic mineral dust samples (from Africa, China and America) and fourteen major minerals found in tropospheric mineral dust aerosol.

For all the mineral dust samples (21 in total) examined, $m_w(90\%)/m_0$ was found to range from 0.0011 ($CaCO_3$) to 0.3080 (montmorillonite), and $\theta(90\%)$ varied between 1.26 (dolomite) and 8.63 (TLF dust). When compared to other types of mineral dust, clay minerals and authentic mineral dust samples usually exhibited larger hygroscopicity on a per mass basis. Our work suggested that overall $m_w(90\%)/m_0$ increased with the BET surface area, indicating that on a per mass basis, mineral dust samples with larger surface area would in general have larger capacities to adsorb water. Our results revealed no dependence of $m_w(90\%)/m_0$ on the amount of soluble materials contained, or no dependence of $\theta(90\%)$ on particle size. In addition, it was found in our work that the Frenkel-Halsey-Hill (FHH) adsorption isotherm could well describe surface coverages of adsorbed water as a function of RH for all the mineral dust investigated, and $A_{FHH}$ and $B_{FHH}$ parameters were determined to be in the range of 0.15-4.39 and 1.10-1.91, respectively.

**Data availability.** Data used in this paper can be found in the main text or supplement of this manuscript.

**Competing interests.** The authors declare that they have no conflict of interest.



**Author contribution.** Mingjin Tang conceived this work; Lanxiadi Chen, Chao Peng, Wenjun Gu, Hanjing Fu and Huanhuan Zhang carried out experiments under the advice of Xing Jian and Mingjin Tang; Lanxiadi Chen, Chao Peng, Athanasios Nenes and Mingjin Tang analyzed the data and wrote the manuscript with input from all the coauthors.

**Financial support.** This work was sponsored by National Natural Science Foundation of China (91744204 and 91644106), Chinese Academy of Sciences (132744KYSB20160036), State Key Laboratory of Organic Geochemistry (SKLOG2016-A05), Guangdong Foundation for Program of Science and Technology Research (2017B030314057 and 2019B121205006), Guangdong Province (2017GC010501) and the CAS Pioneer Hundred Talents program.

**Acknowledgement.** We would like to thank John Crowley (Max Planck Institute for Chemistry, Germany) for providing Saharan dust, Pingqing Fu (Tianjin University, China) for providing TLF dust, and Manolis Romanias (Université Lille, France) for providing M'Bour dust and Bordj dust.

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
