# Peer review of "On mineral dust aerosol hygroscopicity"

_Atmospheric Chemistry and Physics, 2020_

## Referee Comment (RC1) · Anonymous Referee #1 · 15 Jun 2020

The manuscript reports the hygroscopic properties of 7 authentic mineral dust samples and 14 major minerals contained in mineral dust. The manuscript aims to compare the results from this study to previous measurements on the same materials using a number of direct and indirect techniques.

Abstract line 28: replace 'very sensitive balance' with actual name of instrument

Abstract line 32: surface coverage of water has units. The numbers in this line do not have units. They refer to another quantity that should be labeled properly.

Table 1: reports average particle diameter, yet no uncertainty values are listed and the number of significant figures is just not realistic. Revise for accuracy.

Figure 2 is shown in the experimental section when it better moved to the results section

[Figure]

Equation 1, line 189: represents the "fractional" surface coverage, hence, it is unitless. This definition should be emphasized throughout the manuscript. The unit for 'surface area each adsorbed water molecule' is wrong, and the value assumed is too small. There have been measurements of this number for metal oxides that should be used and get cited.

Table 2-5: the formatting of the numbers is better in scientific than normal.

The discussion section 3.2.7 that compares the results to previous work is qualitative in nature. Given the limited novelty of this manuscript, the comparison has to be quantitative based on careful statistical analysis.

It is stated in the manuscript that the experiments were repeated three times (page 9, line 169), yet none of the graphs have error bars on the data points?! This has to be fixed.

Section 3.3 also talks about 'goodness of fit' in qualitative manner. This has to be fixed and the 'goodness of fit' argument has to be based on quantitative analysis.

One important and crucial reason for the differences observed among different studies is the sample pre-treatment prior to water uptake studies. Gas phase water uptake on mineral dust and metal oxides is a surface process and hence the chemical composition of the surface plays an important role in the extent of water surface coverage. Therefore, a detailed discussion on this factor has be added to the manuscript.

The manuscript requires careful reading for grammar, punctuation, and sentence structure.
* * *

---

## Referee Comment (RC2) · Anonymous Referee #2 · 22 Jun 2020

General comments:

The authors present the experimental water adsorption data for 21 mineral dust samples, including 7 authentic mineral dust samples and 14 atmospherically relevant compounds. The motivation of the work is to improve our knowledge of hygroscopicity of mineral dust aerosol and reduce uncertainty of its hygroscopic parameters. The manuscript is well written and I recommend this manuscript to be published in ACP.

Specific comments:

1. Figures S1-S7 display very wide size distribution (often bimodal) of mineral dust particles. So the only average particle diameter presented in table 1 is not enough to characterize particles size in dust samples. Some parameter describing the distribution width (variance, uncertainty, quantiles, etc) should be added.

2. The measurement errors information should be presented in section 2.3 (accuracy of RH and mass measurements, temperature stability).

3 The mass stability criterion reporting in line 169 is doubtful. It is well known that achieving adsorption equilibrium may require several hours, especially at high RH. The figure 2 shows that in some measuring steps (blue line) the stability apparently has not yet been achieved. Reduction of measurement time may result in underestimation of water uptake.

4. In line 499 "Figure 4b" seems to be replaced "Figure 11b". As reported in comment 1 the average diameter is not an informative parameter for the considered dust samples; therefore, the absence of size dependence of surface coverages is not justified.

5. A more detailed explanation of the discrepancy between the results of this work and previous studies is desirable. Especially this concerns the hygroscopicity parameterizations. The values of AFHH differ by more than 10 times for CaCO3 and 6 times for SiO2 although the error of each coefficient is very small (table 6).

---

## Author Comment (AC1) · 29 Sep 2020

Comments from the editorial office and referees are in blue.

Our replies are in black.

Changes to the manuscript are highlighted in red both here and in the revised manuscript.

**Reply to referee #1**

Recommendation:

The manuscript reports the hygroscopic properties of 7 authentic mineral dust samples and 14 major minerals contained in mineral dust. The manuscript aims to compare the results from this study to previous measurements on the same materials using a number of direct and indirect techniques.

**Reply:** We would like to thank ref #1 for reviewing our manuscript, and his/her comments, which helped us largely improve our manuscript, have been carefully addressed in our revision, as detailed below.

Abstract line 28: replace 'very sensitive balance' with actual name of instrument 'vapor sorption analyzer'.

**Reply:** As suggested, in the revised manuscript (page 2), we have changed "using a very sensitive balance" to "using a vapor sorption analyzer".

Abstract line 32: surface coverage of water has units. The numbers in this line do not have units. They refer to another quantity that should be labeled properly.

**Reply:** To be more accurate, "surface coverage" used in our manuscript should be "fractional surface coverage" instead. In the revised manuscript (page 2) we have changed "The surface coverages of adsorbed water" to "The fractional surface coverages of adsorbed water…"

Table 1: reports average particle diameter, yet no uncertainty values are listed and the number of significant figures is just not realistic. Revise for accuracy.

**Reply:** As suggested, in the revised manuscript (page 6) we have added uncertainty values for average particle diameters in Table 1.

Figure 2 is shown in the experimental section when it better moved to the results section

**Reply:** Indeed Figure 2 can be placed in the results section. However, Figure 2 is to show the experimental procedures, and thus we would like to keep Figure 2 in the experimental section.

Equation 1, line 189: represents the "fractional" surface coverage, hence, it is unitless. This definition should be emphasized throughout the manuscript. The unit for 'surface area each adsorbed water molecule' is wrong, and the value assumed is too small. There have been measurements of this number for metal oxides that should be used and get cited.

**Reply:** We agree that we should use "fractional surface coverage". In the revised manuscript (page 10) we have changed "…surface coverage…" to "…fractional surface coverage (abbreviated as surface coverage)…"

The unit for "surface area each adsorbed water molecule" should be "$cm^2$" instead of "$cm^{-2}$", and we have corrected in the revised manuscript (page 10). However, the value we used is correct, and in the revised manuscript (page 10) we have included a few references for this value.

Table 2-5: the formatting of the numbers is better in scientific than normal.

**Reply:** As suggested, we have change the format of these numbers in Tables 2-5, in order to increase their readability.

The discussion section 3.2.7 that compares the results to previous work is qualitative in nature. Given the limited novelty of this manuscript, the comparison has to be quantitative based on careful statistical analysis.

**Reply:** We have already compared our results with those reported in previous studies quantitatively in Sections 3.2.1-3.2.6; therefore, in Section 3.2.7 we only summarize the comparison between our and previous work in a qualitative manner in order to get some general conclusion.

It is stated in the manuscript that the experiments were repeated three times (page 9, line 169), yet none of the graphs have error bars on the data points?! This has to be fixed.

**Reply:** That is a very good point, and as suggested, in the revised manuscript we have included error bars for these figures. It should be noted that some error bars are not clearly visible because the uncertainties are very small.

Section 3.3 also talks about 'goodness of fit' in qualitative manner. This has to be fixed and the 'goodness of fit' argument has to be based on quantitative analysis.

**Reply:** In response to this comment, we have included $R^2$ values in the revised manuscript (page 28):"…can well fit $\theta$ versus RH for all the 21 mineral dust samples examined ($R^2$ were found to be in the range of 0.94-0.99), and the generated…"

One important and crucial reason for the differences observed among different studies is the sample pre-treatment prior to water uptake studies. Gas phase water uptake on mineral dust and metal oxides is a surface process and hence the chemical composition of the surface plays an important role in the extent of water surface coverage. Therefore, a detailed discussion on this factor has be added to the manuscript.

**Reply:** Indeed pretreatment of mineral dust samples may have large effects on their hygroscopic properties. Nevertheless, pretreatment procedures used could differ significantly,

and it is difficult to assess their impacts of hygroscopic properties in different studies. We have added one sentence in the revised manuscript (page 27) to discuss the effect of pretreatment: "For the same dust (at least with the same name), different samples with distinctive hygroscopicity may have been used in our work and previous studies, contributing to the observed discrepancies. In addition, previous work may adopt various pretreatment procedures, and it is difficult to assess the effects of these pretreatment on dust hygroscopicity reported in different studies."

This is also why in the original manuscript (page 27, line 423-429) we have suggested an inter-comparison study in which same samples would be used. In the revised manuscript (page 27) we have expanded it to highlight the pretreatment issue: "To further understand and resolve the discrepancies identified, it will be very useful to distribute the same samples to different groups (in which different techniques would be applied to study their hygroscopic properties) and compare the results obtained; furthermore, these samples should be pretreated with same or very similar protocols after received by different groups."

The manuscript requires careful reading for grammar, punctuation, and sentence structure.

**Reply:** We have carefully read the manuscript again to minimize these errors.

---

## Author Comment (AC2) · 29 Sep 2020

Comments from the editorial office and referees are in blue.

Our replies are in black.

Changes to the manuscript are highlighted in red both here and in the revised manuscript.

**Reply to referee #2**

**Recommendation:**

The authors present the experimental water adsorption data for 21 mineral dust samples, including 7 authentic mineral dust samples and 14 atmospherically relevant compounds. The motivation of the work is to improve our knowledge of hygroscopicity of mineral dust aerosol and reduce uncertainty of its hygroscopic parameters. The manuscript is well written and I recommend this manuscript to be published in ACP.

**Reply:** We would like to thank ref #2 for the very supportive review of our manuscript. His/her comments, which helped us largely improve our manuscript, have been carefully addressed in our revision, as detailed below.

Figures S1-S7 display very wide size distribution (often bimodal) of mineral dust particles. So the only average particle diameter presented in table 1 is not enough to characterize particles size in dust samples. Some parameter describing the distribution width (variance, uncertainty, quantiles, etc) should be added.

**Reply:** As suggested, in the revised manuscript (page 6) we have included standard deviations for average particle diameters in Table 1.

The measurement errors information should be presented in section 2.3 (accuracy of RH and mass measurements, temperature stability).

**Reply:** As suggested, in the revised manuscript (page 9) we have included a sentence to provide these information: "The sample mass could be measured with an accuracy of  $\pm 0.1 \,\mu g$ , and the uncertainties for temperature and RH were  $\pm 0.1 \,^{\circ}C$  and  $\pm 1\%$ ."

The mass stability criterion reporting in line 169 is doubtful. It is well known that achieving adsorption equilibrium may require several hours, especially at high RH. The figure 2 shows that in some measuring steps (blue line) the stability apparently has not yet been achieved. Reduction of measurement time may result in underestimation of water uptake.

**Reply:** Indeed it could take several hours to reach adsorption equilibrium in our experiments. In some of our measurements we also set stability criterion to "mass change was <0.05% in 60 min", and no difference in results was found for the two criterions. In the revised

manuscript (page 9) we have added one sentence to clarify this issue: "...the sample mass was considered to be stable when the mass change was <0.05% in 30 min. In some experiments the sample was considered to reach the equilibrium only when the mass change was <0.05% in 60 min, and no significant difference in results was found for the two equilibrium criterions." In line 499 "Figure 4b" seems to be replaced "Figure 11b". As reported in comment 1 the average diameter is not an informative parameter for the considered dust samples; therefore, the absence of size dependence of surface coverages is not justified.

**Reply:** The referee is correct, and we have changed "Figure 4b" to "Figure 11b" in the revised manuscript (page 32). We also agree with the referee that our dust samples exhibit a wide size distribution, and thus our conclusion on size dependence should be treated with caution. In the revised manuscript (page 32) we have added one sentence to further clarify it: "This conclusion should be used with caution since dust samples used in our work were far from being monodisperse (see Figures S1-S7)."

A more detailed explanation of the discrepancy between the results of this work and previous studies is desirable. Especially this concerns the hygroscopicity parameterizations. The values of  $A_{\text{FHH}}$  differ by more than 10 times for CaCO3 and 6 times for SiO2 although the error of each coefficient is very small (table 6).

**Reply:** In the revised manuscript (page 29-30) we have included one sentence to further explain the difference  $A_{\text{FHH}}$  values measured by different studies: "...while larger differences were observed for  $A_{\text{FHH}}$  values, especially between our study and the work by Kumar et al. (2011a) for SiO2 and CaCO3. One reason for such large difference is that Kumar et al. (2011a) carried out their CCN activity measurements at >100% RH whereas our work on hygroscopic growth was conducted at <100% RH."